# A small step to discover candidate biological control agents from preexisting bioresources by using novel nonribosomal peptide synthetases hidden in activated sludge metagenomes

**Shun Tomita**⬤*, **Kyohei Kuroda**⬤, **Takashi Narihiro***

Bioproduction Research Institute, National Institute of Advanced Industrial Science and Technology (AIST), Sapporo, Hokkaido, Japan

* tomita.s@aist.go.jp (ST); t.narihiro@aist.go.jp (TN)

## Abstract

Biological control agents (BCAs), beneficial organisms that reduce the incidence or severity of plant disease, have been expected to be alternatives to replace chemical pesticides worldwide. To date, BCAs have been screened by culture-dependent methods from various environments. However, previously unknown BCA candidates may be buried and over-looked because this approach preferentially selects only easy-to-culture microbial lineages. To overcome this limitation, as a small-scale test case, we attempted to explore novel BCA candidates by employing the shotgun metagenomic information of the activated sludge (AS) microbiome, which is thought to contain unutilized biological resources. We first performed genome-resolved metagenomics for AS taken from a municipal sewage treatment plant and obtained 97 nonribosomal peptide synthetase (NRPS)/polyketide synthase (PKS)-related gene sequences from 43 metagenomic assembled bins, most of which were assigned to the phyla Proteobacteria and Myxococcota. Furthermore, these NRPS/PKS-related genes are predicted to be novel because they were genetically dissimilar to known NRPS/PKS gene clusters. Of these, the condensation domain of the syringomycin-related NRPS gene cluster was detected in *Rhodoferax*- and *Rhodocyclaceae*-related bins, and its homolog was found in previously reported AS metagenomes as well as the genomes of three strains available from the microbial culture collections, implying their potential BCA ability. Then, we tested the antimicrobial activity of these strains against phytopathogenic fungi to investigate the potential ability of BCA by *in vitro* cultivation and successfully confirmed the actual antifungal activity of three strains harboring a possibly novel NRPS gene cluster. Our findings provide a possible strategy for discovering novel BCAs buried in the environment using genome-resolved metagenomics.

**Data Availability Statement:** All shotgun metagenomic sequence data are publicly available

under the DDBJ Sequence Read Archive database (DRA015582).

**Funding:** This research was funded by the Cabinet Office, Government of Japan, Cross-ministerial Strategic Innovation Promotion Program (SIP), "Technologies for creating next-generation agriculture, forestry and fisheries" (Grant number:18088041, funding agency: Bio-oriented Technology Research Advancement Institution, NARO) and JSPS KAKENHI(Grant Number: JP23K13959). The funders had no role in study design, data collection and analysis, decision to publish, or preparation of the manuscript.

**Competing interests:** The authors have declared that no competing interests exist.

## Introduction

Crop diseases have been a severe problem and have remained a concern for farmers worldwide for many years. Chemical pesticides have helped to reduce crop loss caused by several plant diseases; however, overuse of such chemical pesticides has caused the rapid emergence of pesticide resistance, environmental pollution, and harmful effects on nontarget organisms [1]. Hence, alternative environmentally friendly strategies are necessary for the sustainable development of crop production. In recent years, biological control agents (BCAs), living organisms that reduce the incidence or severity of plant disease, have been expected to be alternatives to replace chemical pesticides worldwide [2, 3]. The increase in the global market size of BCA, including bacteria, fungi, and viruses, may support the exploration and commercialization of novel BCA, *i.e.*, 4.27 billion USD in 2019, and it is expected to reach 11.81 billion USD by 2027, with an annual growth rate of 14.27% [4].

Nonribosomal peptides (NRPs) are known to play a vital role in biological control activity and are synthesized by nonribosomal peptide synthetase (NRPS), which forms either modular (Type I)- or nonmodular (Type II)-gene assemblies [2, 5, 6]. Type I NRPSs consist of large multidomain enzymes that catalyze the formation of natural products via reactions leading to the assembly of both proteinogenic and nonproteinogenic amino acids in a modular manner. A Type I NRPS module mainly consists of three domains: an adenylation (A), a thiolation (T), and a condensation (C) domain [7, 8]. Type II NRPSs lack modular organization, while the catalytic domains are mostly encoded as separate proteins. Hybridization of NRPS and polyketide synthase (PKS) have also been reported [9, 10]. To date, NRP-producing BCAs have been screened by culture-dependent methods from several environments, such as soil, plants, and compost, using general-purpose media that contain rich sources of carbon, energy, salts, amino acids, and vitamins [11–13]. However, previously unknown BCA candidates that produce novel NRPs may be buried and overlooked because this approach preferentially selects only specific microbial lineages. In fact, among 464 NRPs indexed in the NORINE database, 134, 123, and 115 are produced by *Pseudomonas*, *Bacillus*, and *Streptomyces* strains, respectively, accounting for 80.2% of the total [14]. These major bacterial BCA strains are classified into the phyla *Proteobacteria*, *Firmicutes*, and *Actinobacteria*, which are known for their relative ease of culture handling [15, 16]. Although many of such easy-to-culture bacterial strains of the phyla *Proteobacteria*, *Firmicutes*, and *Actinobacteria* other than members of *Pseudomonas*, *Bacillus*, and *Streptomyces* are preserved in culture collections worldwide (e.g., ATCC, DSMZ, JCM, NBRC, etc.), these have rarely been used because it is time consuming to discover novel biological functions, including BCA, among a large number of strains using a culture-dependent approach. These facts imply that a more comprehensive approach is needed to search for novel candidates for BCA from available microbial resources.

The culture-independent metagenomic approach with high-throughput sequencing technology has grown rapidly to solve a part of the abovementioned hurdle of culture-dependent techniques. For instance, several studies generated metagenome assembled genomes (MAGs) to recover complete NRPS and PKS biosynthesis-related gene clusters (BGCs) from a highly complex microbial ecosystem and reported great potential for the discovery of novel compounds with biocontrol activity [17–19]. However, even if novel NRPS-related gene clusters are found in MAGs, it is still difficult to either isolate BCA candidates by a culture-dependent approach or produce physiologically active compounds by a heterologous expression system. To overcome these limitations of both culture-dependent and culture-independent approaches, we propose an indirect approach to efficiently discover BCAs with novel NRPS and NRPS/PKS hybrid gene clusters from culture collections where many underutilized microbial resources are expected to be present. In this study, as a test case, we first performed genome-resolved metagenomics for

activated sludge (AS) taken from municipal sewage treatment plants. The AS process is a biological treatment technology that consists of phylogenetically diverse microorganisms with metabolic activities that remove most pollutants and nutrients in wastewater streams [20]. Bacterial communities in AS are highly diverse and harbor unique microbiomes distinctly different from other habitats, including soil, ocean, human and animal feces, and freshwater [21, 22], and are supposed to be one of the rich and novel microbial resources of BGCs [19, 23]. Such AS-derived metagenomic information is likely to be the key to discovering novel BCAs among the myriad of isolated strains in culture collection. Then, we selected microbial isolates preserved in the culture collection harboring AS-associated NRPS/PKS genes within their genome information. Accessing the resources in culture collection is expected to reduce the labor involved in culturing and isolating target microorganisms from environmental samples. Finally, we tested their antimicrobial activity against phytopathogenic fungi to investigate the potential ability of BCA by *in vitro* cultivation. Our metagenome-informed approach could promote the effective use of available but untouched microbial resources worldwide.

## Materials and methods

### Activated sludge sampling and DNA extraction

A total of 6 AS samples were collected from full-scale conventional activated sludge tanks treating municipal sewage wastewater in the Kyushu region, Japan, with the permission of the administrator when necessary. AS samples were centrifuged at 15,000×*g* for 15 min, and the precipitates were collected. The precipitated AS samples were stored at -80°C prior to DNA extraction. The FastDNA SPIN Kit for Soil (MP Biomedicals) was used to extract total DNA from AS samples following the manufacturer's protocols. A Qubit2.0 Fluorometer (Life Technologies, Carlsbad, CA, USA) was used to determine the DNA purity and concentrations.

### Metagenomic shotgun sequencing, assembly, and binning

Metagenomic shotgun sequencing and data analyses were performed as previously described [24, 25]. Briefly, extracted DNA was applied for shotgun sequencing to generate high-accuracy reads in 2 ×150 mode on an Illumina NovaSeq6000 sequencer (Illumina, CA, USA). The raw reads were trimmed using Trimmomatic v.0.36 [26] and digitally normalized using BBnorm in BBtools v.37.85 (https://sourceforge.net/projects/bbmap). MEGAHIT v.1.2.9 [27] was used to assemble metagenomic reads. The binning of the assembled contigs with >2,000 bp was performed using MetaBAT v.2.12.1 [28]. The completeness and contamination of each bin were checked using Check M v.1.0.11 [29]. The taxonomical classification of each bin was estimated using GTDBtk v.1.4.1 (GTDB release95; default parameters) [30].

### Screening secondary metabolism genes and phylogenetic analysis of NRPS domains

For identification of the secondary metabolism genes, nucleotide sequences of contigs for a total of 180 bins were submitted to locally installed Anti-SMASH software (v.5.1.2, with the options–clusterblast–smcogs–limit 1500) [31]. To search for novel candidates for BCA from available microbial resources, all of the C domain sequences were classified and trimmed by the NaPDoS (Natural Product Domain Seeker) web server [32]. To complement NRPS not included in the NaPDoS database and to accurately identify microbial isolates closely related to metagenome-derived NRPS, the C domain sequences were submitted to BLASTP against the RefSeq database [33] using the default parameters. Then, the five best hits of each C domain sequence were clustered at ≥ 40% amino acid similarity using CD-HIT v.4.8.1 [34]

and used for the construction of a maximum likelihood tree to show the whole phylogenetic diversity of the C domains of NRPS detected from the AS microbial communities. For a detailed analysis of the C domain sequences of Burkholderiales-related bins 43 and 137, the best 10 hits of the RefSeq database and previously reported environmental sequences were used for the construction of a maximum likelihood tree. The resulting alignments and trees were exported, and then the trees were manually checked and annotated. The complete genome sequences of *Pseudoduganella ginsengisoli* JCM 30745 (accession no. GCA_009720865.1), *Noviherbaspirillum denitrificans* JCM 17722 (GCA_002211445.1), and *Rhodoferax koreense* JCM 31441 (GCA_001955695.1), which were all genomic data of the strains available in the GenBank database at the time of analysis (December 2021), were obtained for comparison of gene cassettes with bins 43 and 137. The functional annotation of the gene cassettes was carried out by JGI IMG/M-ER pipelines [35].

## Fungal and bacterial strains and growth conditions

*Botrytis cinerea* Persoon MAFF 306935, *Verticillium dahliae* MAFF 236075, and *Colletotrichum plurivorum* MAFF 306007, the phytopathogenic fungi of tomato, were prepared on potato dextrose agar (PDA) plates (Difco) at 25°C for 14 days. After incubation on the plates, the conidia were rinsed from the surface of the plate with 0.5% Tween 80. This suspension was supplemented with 25% glycerol and placed in a −80°C freezer for storage until use. *R. koreense* JCM 31441, *Rhodoferax sediminis* JCM 32677, *N. denitrificans* JCM 17722, and *P. ginsengisoli* JCM 30745 were provided by the RIKEN BRC through the National BioResource Project of the MEXT, Japan. *Bacillus subtilis* NBRC 109107 was provided by the Biological Resource Center, National Institute of Technology and Evaluation. These strains were cultivated in R2A medium (Daigo) at 25°C and kept as glycerol stocks at -80°C until use.

## Antifungal activity test *in vitro* against phytopathogenic fungi

Each antagonistic bacterial strain was inoculated in a straight line at a distance of 2.25 cm from the center on a 9 cm R2A agar plate diameter with an inoculating loop. Fungal strains were cultured on PDA and maintained at 25°C in the dark. A 14-day-old mycelial plug of the fungal strain was cut out using a 5 mm diameter cork borer and placed at the center of the plate. The plates were sealed with parafilm, and after incubation at 25°C for six days, the inhibitory zones were observed. Triplicate experiments were performed. The inhibitory effect of antagonistic bacteria on the fungal pathogen was calculated. Three or four biological replicates were performed, and an average was taken. Inhibition rate (%) = (diameter of a colony far from antagonistic bacterium-diameter of colony close to antagonistic bacterium)/diameter of colony far from antagonistic bacterium × 100. The data were analyzed by one-way ANOVA followed by Tukey's honestly significant difference post hoc test with the freely available statistical analysis program tool js-STAR (http://www.kisnet. or.jp/nappa/software/star/), and differences were considered to be significant at $p < 0.05$.

## Deposition of DNA sequence data

The raw sequence and binned metagenome data were deposited into the DDBJ Sequence Read Archive database (DRA015582).

## Results and discussion

### Draft genome reconstruction from activated sludge

We successfully obtained 180 high-quality (*i.e.*, >80% completeness, <10% contamination) draft genomes (hereafter called "bins") from shotgun metagenomic sequence data. S1 Table

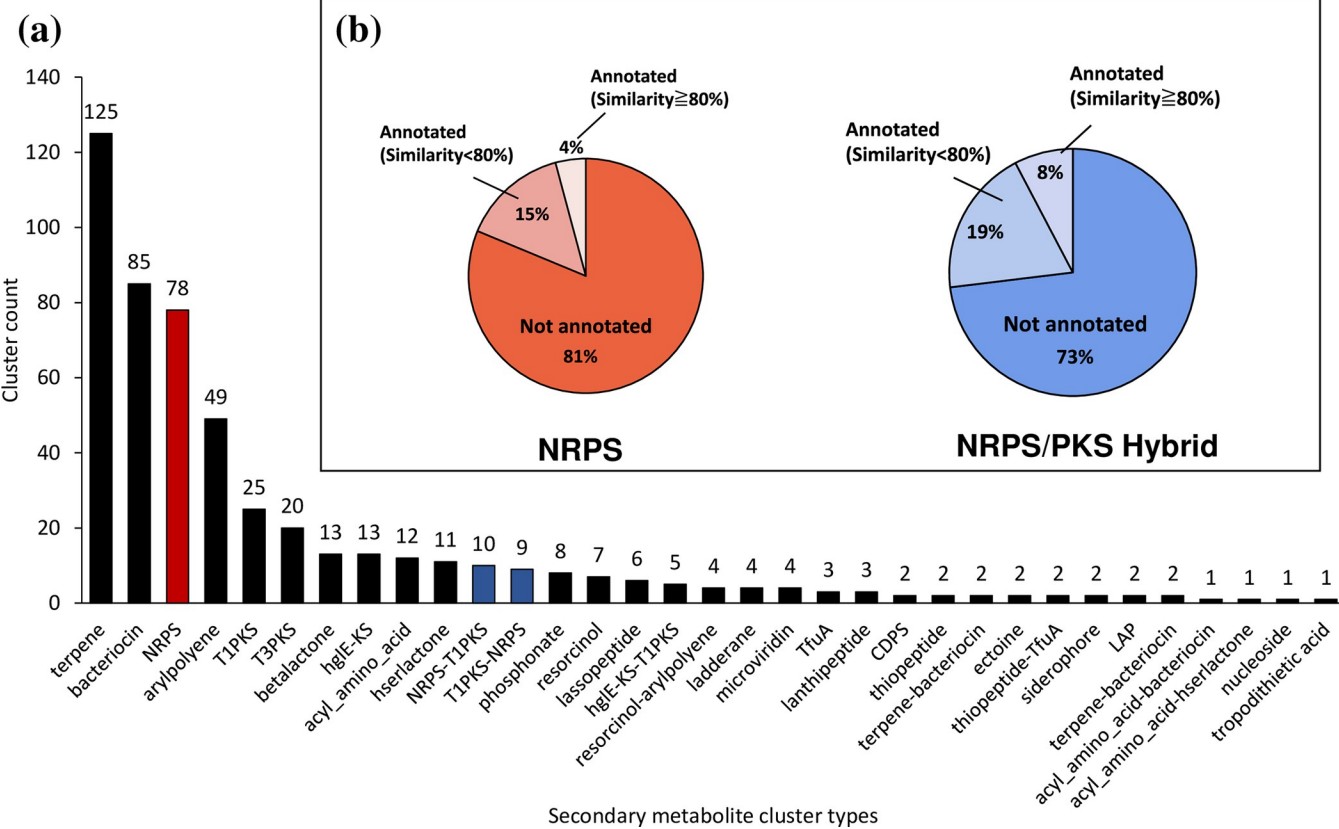

**Fig 1. Secondary metabolite clusters obtained by Anti-SMASH.** (a) Abundance of secondary metabolite gene cluster types in the 180 bins recovered from activated sludge obtained with Anti-SMASH. The red bar indicates the NRPS. The blue bar indicates NRPS/PKS hybrid clusters (NRPS-PKS and PKS-NRPS). (b) Comparisons with known NRPS and NRPS/PKS hybrid clusters from the MIBiG dataset in Anti-SMASH.

shows the general information about each bin, including completeness, genome size, number of open reading frames (ORFs), and taxonomical classifications according to GTDBtk phylogeny. By using Anti-SMASH software, at least one gene cluster associated with secondary metabolite biosynthesis was found in 97 bins, totaling 514 gene clusters (Fig 1A). The most abundant gene cluster was categorized into terpene biosynthesis (24.3%), followed by bacteriocin biosynthesis (16.6%) and then NRPS (15.2%). Terpene biosynthesis gene clusters are well known in the genomes of many plants and fungi but have recently been found to be distributed in bacterial genomes [36, 37]. For example, 262 terpene synthases were observed in bacterial draft genomes, and 13 were determined to be involved in the production of previously unidentified sesquiterpenes and diterpenes [38]. Bacteriocins are a large family of ribosomally synthesized antimicrobial peptides that have antimicrobial activity against strains of closely related species [39]. In addition, unlike antibiotics, bacteriocins are sensitive to proteases but harmless to the human body and surrounding environment [40, 41]. Although some studies have been conducted to discover bacteriocin gene clusters by metagenomic approaches, most of these are detected in the host-associated microbiome [42] or fermented food microbiome [43]. Consequently, our results suggest that the bacterial draft genome from AS might be a fertile source of novel terpene and bacteriocin compounds. A total of 78 NRPS (15.2%) and 19 hybrid NRPS/PKS (3.6%) clusters were observed in 43 bins (Fig 1A). The complete modules containing A, T, and C domains were found in 22 clusters (22.45% of the total) (Table 1), among which are 14 with one complete set of modules, 7 with two sets, and 1 with three sets. A previous study

**Table 1. The number of clusters of NRPS and NRPS/PKS hybrids harbored by each bin.**

| Bin ID | Class | Order | Family | Genus | NRPS | NRPS/PKS Hybrid | Total | NRPS and NRPS/PKS Hybrid with complete modules |
|---|---|---|---|---|---|---|---|---|
| **Phylum Proteobacteria** | | | | | | | | |
| 533 | Alphaproteobacteria | Rhodobacterales | Rhodobacteraceae | Tabrizicola | 0 | 1 | 1 | 0 |
| 424 | | | | unidentified | 1 | 0 | 1 | 0 |
| 536 | | Sphingomonadales | Sphingomonadaceae | Novosphingobium | 1 | 0 | 1 | 0 |
| 43 | | | Burkholderiaceae | Rhodoferax | 1 | 0 | 1 | 1 |
| 167 | | | | Rhodoferax | 1 | 0 | 1 | 0 |
| 454 | Gammaproteobacteria | Burkholderiales | Rhodocyclaceae | Accumulibacter | 1 | 0 | 1 | 0 |
| 189 | | | | Methyloversatilis | 3 | 0 | 3 | 0 |
| 134 | | | | Sterolibacterium | 5 | 1 | 6 | 3 |
| 137 | | | | unidentified | 1 | 0 | 1 | 1 |
| 199 | | | | unidentified | 2 | 0 | 2 | 0 |
| 266 | | Pseudomonadales | Moraxellaceae | unidentified | 1 | 1 | 2 | 0 |
| 187 | | Steroidobacterales | Steroidobacteraceae | ZC4RG30 | 2 | 1 | 3 | 1 |
| 142 | | | | unidentified | 3 | 1 | 4 | 0 |
| 527 | | Xanthomonadales | Xanthomonadaceae | Aquimonas | 4 | 0 | 4 | 1 |
| **Phylum Myxococcota** | | | | | | | | |
| 392 | UBA796 | UBA796 | GCA-2862545 | unidentified | 2 | 0 | 2 | 0 |
| 212 | UBA9042 | UBA9042 | UBA9042 | unidentified | 1 | 0 | 1 | 0 |
| 62 | | PHBI01 | PHBI01 | unidentified | 3 | 0 | 3 | 0 |
| 459 | Polyangia | Nannocystales | Nannocystaceae | Nannocystis | 3 | 3 | 6 | 1 |
| 105 | | Palsa-1104 | unidentified | unidentified | 5 | 2 | 7 | 2 |
| 463 | | Polyangiales | Polyangiaceae | Minicystis | 3 | 1 | 4 | 1 |
| 119 | | | Polyangiaceae | unidentified | 1 | 0 | 1 | 0 |
| 286 | | | | unidentified | 5 | 0 | 5 | 0 |
| 41 | | | Ga0077539 | Ga0077539 | 2 | 0 | 2 | 0 |
| 232 | | | | SCUS01 | 1 | 0 | 1 | 0 |
| 389 | | | | unidentified | 2 | 1 | 3 | 0 |
| 457 | | unidentified | unidentified | unidentified | 2 | 1 | 3 | 0 |
| 468 | | unidentified | unidentified | unidentified | 3 | 0 | 3 | 0 |
| **Phylum Bacteroidota** | | | | | | | | |
| 125 | Bacteroidia | AKYH767 | UBA4408 | unidentified | 1 | 0 | 1 | 0 |
| 67 | | | unidentified | unidentified | 1 | 1 | 2 | 1 |
| 344 | | Chitinophagales | Chitinophagaceae | OLB11 | 1 | 0 | 1 | 0 |
| 419 | | | | unidentified | 0 | 1 | 1 | 1 |
| 472 | | Flavobacteriales | PHOS-HE28 | PHOS-HE28 | 2 | 0 | 2 | 1 |
| 305 | | Sphingobacteriales | Sphingobacteriaceae | Daejeonella | 0 | 1 | 1 | 1 |
| **Phylum Verrucomicrobiota** | | | | | | | | |
| 285 | Verrucomicrobiae | Pedosphaerales | UBA9464 | SXSK01 | 2 | 1 | 3 | 1 |
| 114 | | | UBA11320 | UBA11320 | 1 | 1 | 2 | 1 |
| 467 | | | unidentified | unidentified | 2 | 0 | 2 | 2 |
| **Phylum Actinobacteriota** | | | | | | | | |
| 437 | Actinomycetia | Actinomycetales | Dermatophilaceae | Phycicoccus_A | 2 | 0 | 2 | 0 |
| 487 | Acidimicrobiia | Acidimicrobiales | Microtrichaceae | UBA11034 | 1 | 0 | 1 | 0 |

*(Continued)*

**Table 1.** (Continued)

| Bin ID | Class | Order | Family | Genus | NRPS | NRPS/PKS Hybrid | Total | NRPS and NRPS/PKS Hybrid with complete modules |
|---|---|---|---|---|---|---|---|---|
| **Phylum Chloroflexota** | | | | | | | | |
| 282 | Anaerolineae | Anaerolineales | EnvOPS12 | OLB14 | 1 | 1 | 2 | 1 |
| 178 | | | | OLB14 | 1 | 0 | 1 | 1 |
| **Others (phylum; class)** | | | | | | | | |
| 423 | Thermoanaerobaculia | UBA5066 | UBA5066 | UBA5066 | 2 | 0 | 2 | 0 |
| 180 | Eremiobacterota; UBP9 | | UBA4705 | unidentified | 1 | 0 | 1 | 0 |
| 241 | Fibrobacteria | UBA5070 | UBA5070 | UBA5070 | 1 | 0 | 1 | 1 |

"Complete modules" means containing A, T, and C domains.

reported that NRPS and PKS in the AS MAGs appeared to be mostly very short despite being uninterrupted by contig breaks, suggesting that PKs and NRPs in AS produce simple compounds compared with the structure of the known multimodular NRPS and PKS [23]. Here, the detailed analysis with a special emphasis on the NRPS and hybrid NRPS/PKS gene clusters, well-known biological control factors against various plant diseases, is described in the following sections.

## NRPS/PKS-related gene screening

The comparison of the amino acid sequence identity of these NRPS/PKS-related genes retrieved from AS metagenomes to known BGCs was performed by using Anti-SMASH. As expected, 81% of NRPS and 73% of NRPS/PKS hybrid clusters were not annotated as known NRPS/PKS-related genes (Fig 1B), suggesting that novel BCA candidates having NRPS or NRPS/PKS hybrid gene clusters were abundant in the AS metagenomes. For taxonomical classification at the phylum level for the bins harboring NRPS and NRPS/PKS hybrid clusters, the majority of the bins belonged to the phyla Proteobacteria, Myxococcota, Bacteroidota, Verrucomicrobiota, and Actinobacteriota (S1 Fig). The bins associated with the phyla Proteobacteria and Myxococcota encoded both NRPS and NRPS/PKS hybrid gene clusters, with 33.3% (NRPS) and 31.3% (NRPS/PKS hybrid) proportions, respectively (S1 Fig). Table 1 shows the number of clusters of NRPS and NRPS/PKS hybrids harbored by each bin. Importantly, there were no NRPS or NRPS/PKS hybrid gene clusters associated with the well-recognized BCA, such as the genera *Pseudomonas*, *Bacillus*, and *Streptomyces*, indicating that the activated sludge metagenome was rich in underexploited microbial genetic resources. Twenty-five bins contained 43 harbored multiple clusters, with Myxococcota being the most abundant (10 bins). Myxococcota is a phylum of gram-negative bacteria with a specific life cycle that produces various types of secondary metabolites with novel chemical structures and biological activities [44]. Myxococcota-related bins harbored the highest number of NRPS and NRPS/PKS hybrid clusters per draft genome collected from AS metagenomes as abundant as the Proteobacteria phylum. However, Myxococcota has received less attention than commonly reported biological control agents such as *Bacillus*, *Pseudomonas*, and *Streptomyces*. Recently, some studies have screened Myxococcota-related BCA candidates and evaluated their biological control activity against plant pathogens. For example, *Myxococcus xanthus* R31 has potential biocontrol activity against tomato bacterial wilt by inhibiting *Ralstonia solanacearum* [45].

*Citreicoccus inhibens* M34[T] shows antifungal and bacteriolytic activity against several phytopathogenic bacteria [46]. *Corallococcus* sp. EGB controlled cucumber *Fusarium* wilt in a field experiment [47]. All these strains were isolated from soil, and there are no reports on the biocontrol of Myxococcota isolated from AS. Hence, further study will be necessary to cultivate these Myxococcota-related organisms from AS and to ensure their actual biological control activity.

Within the phylum *Proteobacteria*, the genera *Pseudomonas* and *Burkholderia* are known to harbor NRPS and NRPS/PKS hybrid clusters, producing lipopeptides and siderophores, which act as BCA against many types of plant diseases [5, 48]. However, in the AS metagenomes, no bins were identified as *Pseudomonas* and *Burkholderia* members (Table 1). Instead, the metagenomic bins harboring NRPS or NRPS/PKS hybrid clusters were assigned to phylogenetically diverse genera, including *Sterolibacterium*, *Aquimonas*, *Methyloversatilis*, *Accumulibacter*, *Rhodoferax*, *Novosphingobium*, and *Tabrizicola*, suggesting the presence of novel BCA candidates in the AS ecosystem other than the well-known BCAs. These observations have shed light on the biocontrol potential of microorganisms in AS that have not been studied for discovering BCA.

Furthermore, to search for novel candidates for BCA from available microbial resources, we performed C-domain phylogenetic analysis by NaPDoS. C domain phylogeny delineates functional subtypes as opposed to species relationships so that it has been used to assess the diversity and richness of NRPSs in environments and identify new functional classes that might be associated with uncharacterized biosynthetic mechanisms [49]. First, a total of 53 condensation (C) domains were recovered from NRPS and NRPS/PKS hybrid clusters in all bins and submitted to NaPDoS analysis (S2 Table). All the C domains were also submitted to similarity analysis using the BLASTP platform, and the best five hits of each sequence were chosen and used for phylogenetic analyses with NaPDoS. As a result, most C domain sequences of the bins formed separate clusters based on the phylogenetic tree and were only remotely related to C domains of the known sequences in NaPDoS (S2 Fig). Among these bins that formed separate clusters, we found that *Rhodoferax* bin 43 and *Rhodocyclaceae* bin 137 encoded the C domain of syringomycin synthetase with relatively low amino acid similarities compared with NaPDoS reference sequences (31%–33%) (S2 Table). Syringomycin is a well-known lipopeptide secreted by *Pseudomonas syringae* with a broad spectrum of antifungal activities [50]. These two bins have a complete module in the NRPS clusters (Table 1) annotated by Anti-SMASH, so we focused on these two bins having the novel NRPS genes as BCA candidates for further analysis and experiments.

These C domain sequences in bins 43 and 137 formed a clade with those found in previously reported AS metagenomic datasets [19, 51, 52], suggesting that the novel BCAs having the NRPS genes associated with this clade may be occasionally present in the activated sludge environment (Fig 2). In addition to these metagenome-derived sequences, several C domain-like sequences of bacterial strains available in culture collections, including *Pseudoduganella ginsengisoli* JCM 30745, *Noviherbaspirillum denitrificans* JCM 17722, and *Rhodoferax koreense* JCM 31441, were classified in this clade. These strains were originally isolated from soil [53], paddy fields [54], and sludge [55], but there have been no reports on whether they have the same activity as BCA. Fig 3 shows the gene cassettes associated with NRPS clusters in three JCM strains' genomes, *Rhodoferax* bin 43, and *Rhodocyclaceae* bin 137. All of these organisms possess NRPS with 51%–58% amino acid identity to that of bin 43. In addition, two enzymes were found to be associated with the NRPS: one has a soluble N-ethylmaleimide-sensitive factor attachment receptor (SNARE)-associated domain, and the other has a patatin-like phospholipase domain. The details of the annotated genes are shown in S3 Table. SNARE-related enzymes are thought to be used for promoting or blocking membrane fusion and act against

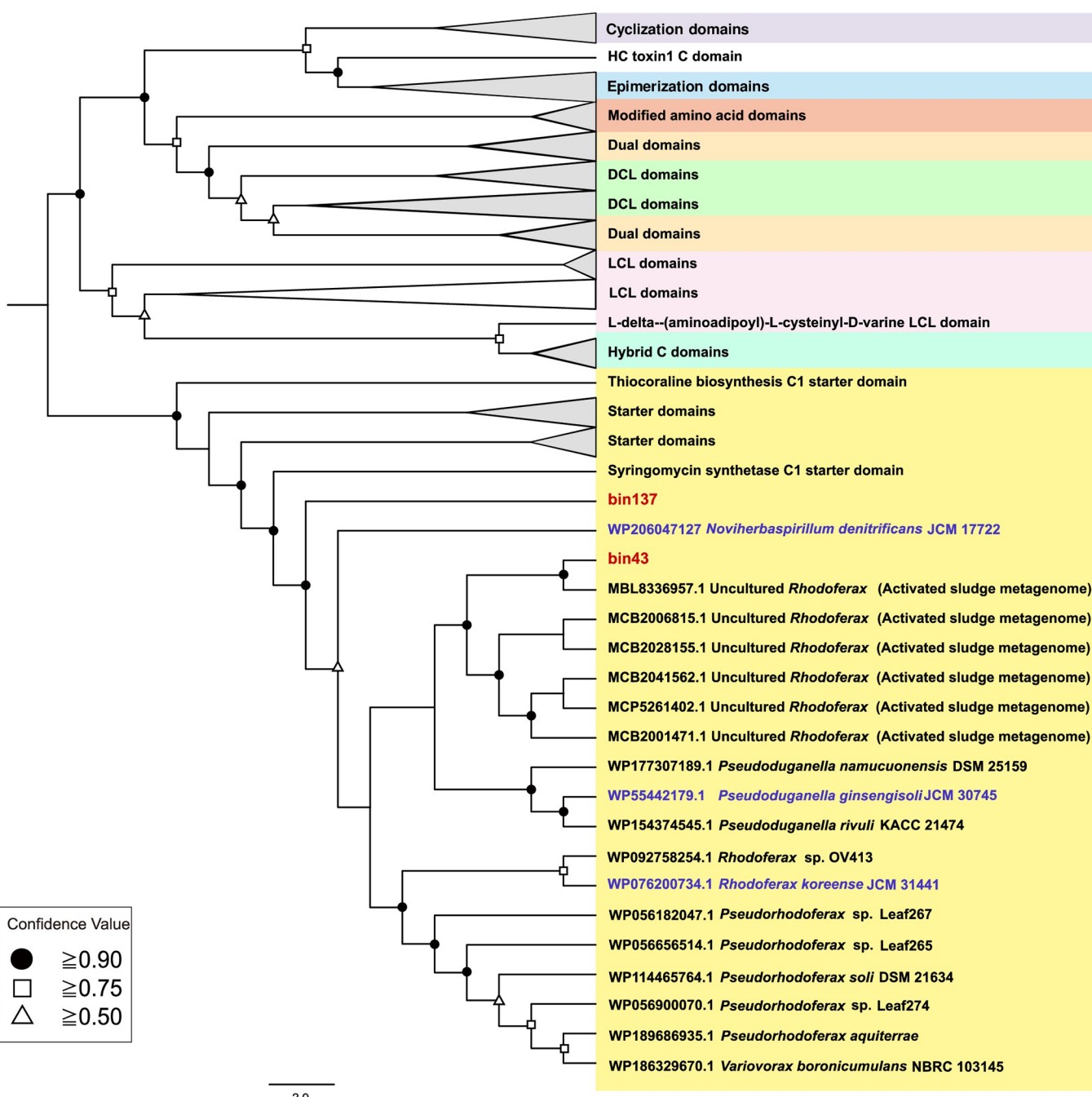

**Fig 2. Phylogenetic tree of C domains (bin 43 and 137, the top 10 BLAST results on RefSeq, the NaPDoS reference sequences and previously reported environmental sequences).** The tree was generated by the NaPDoS pipeline (using FASTTREE and the maximum likelihood algorithm). Dots, squares, and triangles on nodes indicate the confidence value. The sequences from the bins are in red. The sequences of strains used for the antifungal activity test are shown in blue.

eukaryotes [56, 57]. Phospholipases, including patatin, are considered virulence factors for pathogenic bacteria [58, 59]. To estimate whether these NRPS clusters are involved in NRP biosynthesis and whether the strains with these clusters are potential BCAs, we performed further experiments on *in vitro* antifungal activity.

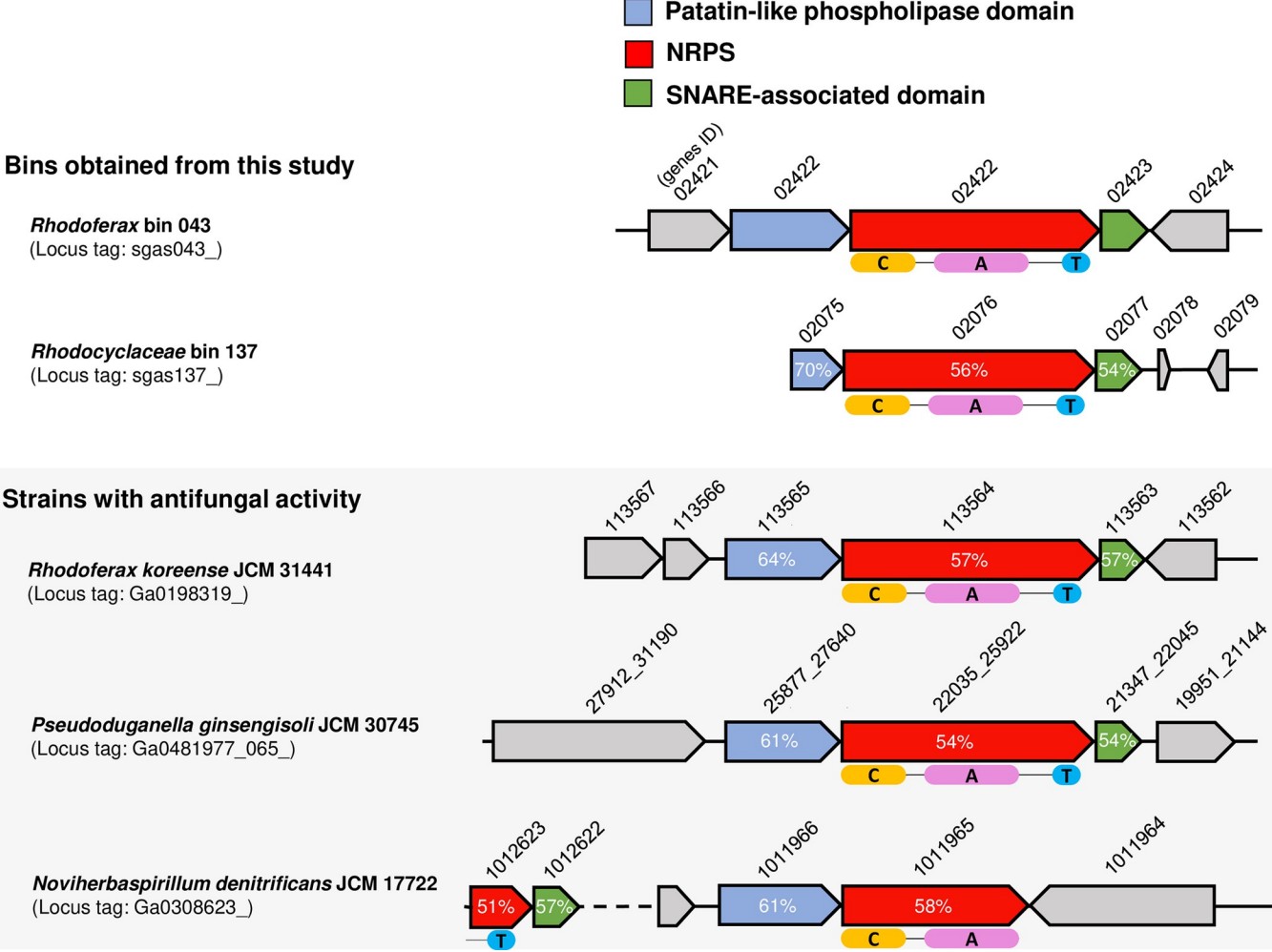

**Fig 3. Comparison of gene cassettes associated with NRPS in bin 43, 137 and *R. koreense* JCM 31441, *P. ginsengisoli* JCM 30745, and *N. denitrificans* JCM 17722.** Numbers in gene boxes indicate amino acid sequence identity (%) to the corresponding gene of bin43 (*Rhodoferax*). NRPS domains are labeled as follows: C condensation, A adenylation, and T thiolation. Each domain in *N. denitrificans* JCM 17722 is located in different NRPS genes; Ga308623_01012623 possesses a T domain, and Ga0308623_1011965 possesses C and A domains.

## Antifungal activity of biocontrol candidates against plant pathogenic fungi

We evaluated the antifungal activity of candidate BCA strains, *P. ginsengisoli* JCM 30745, *N. denitrificans* JCM 17722, and *R. koreense* JCM 31441, whose C domains of syringomycin synthetases are clustered with those of bins 43 and 137. *Rhodoferax sediminis* JCM 32677 was evaluated as a negative control harboring no C domain sequence in the genome. In addition, *Bacillus subtilis* NBRC 109107 was evaluated as a positive control producing cyclic lipopeptide synthesized by NRPS [60]. The antifungal activities were tested *in vitro* against *Botrytis cinerea* MAFF 306935, *Verticillium dahliae* MAFF 236075, and *Colletotrichum plurivorum* MAFF 306007, the causal agents of fungal disease in tomatoes. As a result, only *P. ginsengisoli* JCM 30745 and *Bacillus subtilis* NBRC 109107 had antifungal activity against *V. dahliae*, while *R. koreense* JCM 31144, *N. denitrificans* JCM 7452, and *R. sediminis* JCM 32677 did not; additionally, all tested strains did not show antifungal activity against *C. plurivorum* except *Bacillus subtilis* NBRC 109107 (S3 Fig). Meanwhile, all three candidate BCA strains except for *R. sediminis* JCM 32677 showed antifungal activity against *B. cinerea*; therefore, the antifungal activities were quantified by measuring the

Control | *P. ginsengisoli* JCM 30745 | *N. denitrificans* JCM 17722

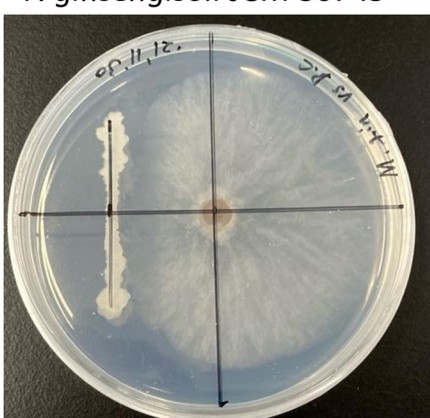
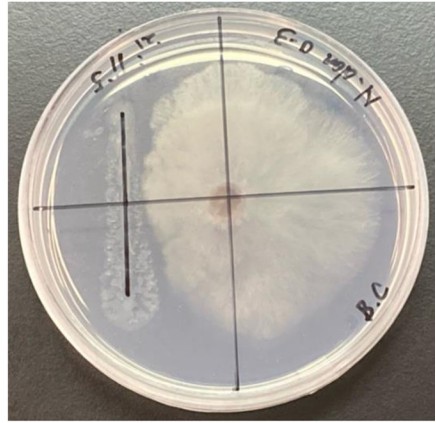

*Bacillus subtilis* NBRC 109107 | *R. koreense* JCM 31441 | *R. sediminis* JCM 32677

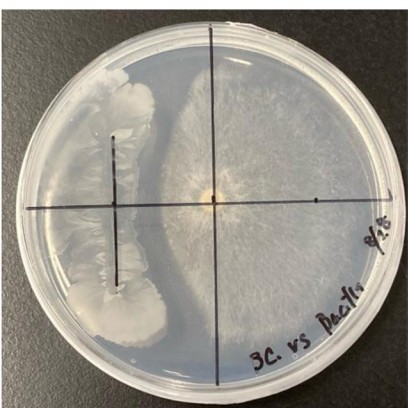
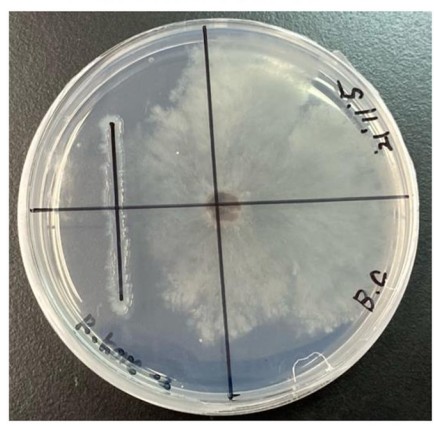
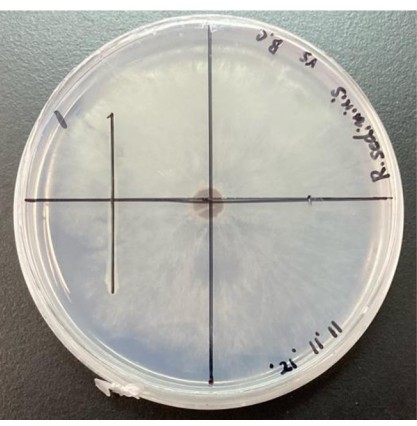

**Fig 4. Antifungal activities of *R. koreense* JCM 31441, *P. ginsengisoli* JCM 30745, *N. denitrificans* JCM 17722, and *R. sediminis* JCM 32677 against *B. cinerea*.** An overnight culture of each bacterial strain was inoculated in a straight line on an R2A agar plate with an inoculating loop. *Rhodoferax sediminis* JCM 32677 was used as a negative control.

diameter of colonies to quantify the antifungal activity (Fig 4). The percent inhibition of mycelial growth by *P. ginsengisoli* JCM 30745, *N. denitrificans* JCM 7452, *R. koreense* JCM 31144 and *Bacillus subtilis* NBRC 109107 was 45.1%, 43.26%, 47.37%, and 58.94%, respectively, which was significantly higher than that of *R. sediminis* JCM 32677 (11.55%) (Table 2). The results indicated

**Table 2. Results of the antifungal activity of BCA candidates with *B. cinerea*.**

| Used bacteria | Inhibition rate (%) |
|---|---|
| *Pseudoduganella ginsengisoli* JCM 30745 | 47.33 ± 1.83 a |
| *Rhodoferax koreense* JCM 31441 | 45.1 ± 8.45 a |
| *Noviherbaspirillum denitrificans* JCM 17722 | 43.26 ± 17.23 a |
| *Bacillus subtilis* NBRC 109107 | 58.94 ± 5.55 a |
| *Rhodoferax sediminis* JCM 32677 | 11.55 ± 7.85 b |

S.E. = Standard Error. Data in the table are expressed as the mean ± S.E. Different letters in the same row indicate significant differences at the $p < 0.05$ level by one-way analysis of variance followed by Tukey's honestly significant difference post hoc test. *Bacillus subtilis* NBRC 109107 was used as a positive control, and *Rhodoferax sediminis* JCM 32677 was used as a negative control.

that the three BCA candidates found in this study have antifungal activity similar to that of *Bacillus subtilis* NBRC 109107, a known NRP-producing strain. Anti-SMASH analysis showed that *R. koreense* JCM 31441 and *N. denitrificans* JCM 17722 do not possess any anti-microbial agent-related genes other than the syringomycin synthetase-related C domain of NRPS (S4 Fig), implying that the C domain of NRPS in these bacterial strains may be involved in the biosynthesis of novel NRPs that inhibit the growth of *B. cinerea*. On the other hand, *P. ginsengisoli* JCM 30745 possesses several BGCs, including NRPS, PKS, and RiPP (ribosomally synthesized and post-translationally modified peptide) [61], which are known antimicrobial agents (S4 Fig), which might explain why *P. ginsengisoli* JCM 30745 showed antifungal activity against *V. dahliae*, while the other two BCA candidate strains did not. We successfully confirmed antifungal activity against phytopathogens with novel candidate strains for BCA from available microbial resources, although it is not yet clear whether the C domain in the strains is related to the activity. In the future, to confirm the exact NRPS products with antifungal activity, it is clearly necessary to investigate in more detail whether this NRPS gene cluster is involved in NRP biosynthesis by constructing a gene disruption mutant or using a heterologous expression system and by identifying the chemical structure using mass spectrometry.

## Conclusion

This study aimed to discover novel BCAs from available microbial resources using metagenomic information of AS microbiota containing underutilized NRPS/PKS-related genes as a test case. We successfully reconstructed the draft genomes from AS samples by a metagenomic approach, which resulted in 43 bins encompassing a total of 78 NRPS and 19 hybrid NRPS/PKS clusters. Most of them were not assigned to known compounds, demonstrating that the AS microbiome has the hidden ability to produce novel compounds contributing to biological control. Furthermore, we successfully confirmed the actual antifungal activity of three strains obtained from available microbial resources harboring possibly novel NRPS genes encoded in Burkholderiales-related bins and previously reported AS-derived metagenomes. Overall, although the present study is a test case employing a small-scale AS metagenomic dataset, it is expected that more comprehensive exploration of BCAs from available microbial resources will be possible in the future using large-scale metagenomic information acquired from various environments and genomic information of cultured isolates.

## Supporting information

**S1 Table. General information of metagenomic bins.**
(XLSX)

**S2 Table. NRPS-related gene (C domain) identified in the bins using NaPDoS.**
(XLSX)

**S3 Table. Detailed annotation of the gene cassettes associated with NRPS clusters in *Rhodoferax* bin 43, *Rhodocyclaceae* bin 137, and three strains' genomes with antifungal activity.**
(XLSX)

**S1 Fig. Taxonomic classification at the phylum level for the bins showing NRPS and NRPS/PKS hybrid clusters.**
(TIF)

**S2 Fig. NaPDoS phylogenetic tree of C domains in bins obtained from AS, the top 5 BLAST results on the RefSeq database, and the NaPDoS reference sequence.**
(TIF)

**S3 Fig. Antifungal activities of *R. koreense* JCM 31441, *P. ginsengisoli* JCM 30745, *N. denitrificans* JCM 17722, *Bacillus subtilis* NBRC 109107, and *R. sediminis* JCM 32677 against *C. plurivorum and V. dahliae.* An overnight culture of each bacterial strain was inoculated in a straight line on an R2A agar plate with an inoculating loop.**
(TIF)

**S4 Fig. Secondary metabolite genes in *R. koreense* JCM 31441, *P. ginsengisoli* JCM 30745, and *N. denitrificans* JCM 17722 obtained with Anti-SMASH.** The red box shows NRPS genes with the C domain of syringomycin synthetase.
(TIF)

**S1 Graphical abstract.**
(PDF)

## Acknowledgments

Sequence data submission was supported by the D-way Submission Portal provided by the DNA Data Bank of Japan (DDBJ).

## Author Contributions

**Conceptualization:** Shun Tomita, Takashi Narihiro.

**Data curation:** Shun Tomita, Kyohei Kuroda.

**Funding acquisition:** Shun Tomita, Kyohei Kuroda, Takashi Narihiro.

**Investigation:** Shun Tomita, Kyohei Kuroda.

**Methodology:** Shun Tomita, Kyohei Kuroda.

**Project administration:** Takashi Narihiro.

**Resources:** Takashi Narihiro.

**Supervision:** Takashi Narihiro.

**Writing – original draft:** Shun Tomita.

**Writing – review & editing:** Kyohei Kuroda, Takashi Narihiro.

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
