## [Decision Letter · Decision Letter 0]

15 Aug 2023

PONE-D-23-16870A small step to discover candidate biological control agents from pre-existing bioresources by using novel nonribosomal peptide synthases hidden in the activated sludge metagenomesPLOS ONE

Dear Dr. Tomita,

Thank you for submitting your manuscript to PLOS ONE. After careful consideration, we feel that it has merit but does not fully meet PLOS ONE’s publication criteria as it currently stands. Therefore, we invite you to submit a revised version of the manuscript that addresses the points raised during the review process.

We look forward to receiving your revised manuscript.

Kind regards,

Laila Adel Ziko

Academic Editor

PLOS ONE

“This work was supported by the Cabinet Office, Government of Japan, Cross-ministerial Strategic Innovation Promotion Program (SIP), “Technologies for creating next-generation agriculture, forestry and fisheries” (funding agency: Bio-oriented Technology Research Advancement Institution, NARO) and partly supported by JSPS KAKENHI, Grant Number JP23K13959. Sequence data submission was supported by the D-way Submission Portal provided by the DNA Data Bank of Japan (DDBJ).”

“This research was funded by the Cabinet Office, Government of Japan, Cross-ministerial Strategic Innovation Promotion Program (SIP), “Technologies for creating next-generation agriculture, forestry and fisheries” (Grant number:18088041, funding agency: Bio-oriented Technology Research Advancement Institution, NARO) and JSPS KAKENHI(Grant Number:JP23K13959). The funders had no role in study design, data collection and analysis, decision to publish, or preparation of the manuscript.”

Additional Editor Comments:

The manuscript is pertaining to a current topic of interest and the field of BGCs is now growing with respect to bioinformatics and genome mining. It is harder to deal definitely to deal with metagenomes and it is thus a challenge. However, the lack of positive control is weak point in the current draft in addition to the product itself not being characterized. Please address all the reviewers’ comments carefully in order to have the manuscript improved. It is saying a strong message but also the wording needs work as well.

Reviewers' comments:

Reviewer's Responses to Questions

**Comments to the Author**

1. Is the manuscript technically sound, and do the data support the conclusions?

Reviewer #1: No

Reviewer #2: Yes

2. Has the statistical analysis been performed appropriately and rigorously? 

Reviewer #1: N/A

Reviewer #2: Yes

3. Have the authors made all data underlying the findings in their manuscript fully available?

Reviewer #1: Yes

Reviewer #2: Yes

4. Is the manuscript presented in an intelligible fashion and written in standard English?

Reviewer #1: Yes

Reviewer #2: Yes

5. Review Comments to the Author

Reviewer #1: The authors described a small-scale test case to select NRPS or NRPS/PKS clusters from activated sludge shotgun metagenome data. Next, they looked at culture collections for strains that harbor the selected clusters, with the end goal to use these strains as biological control agents. Next, they investigated the antifungal activity of these strains. Even though the authors identified three strains that harbor a similar NRPS gene between them, the claim that the product of this NRPS is responsible for the antifungal activity is weak, and further experiments are needed to corroborate with this hypothesis. Moreover, some sections must be re-written to be more clear about how and why the analysis was performed.

Reviewer #2: Biological control agents (BCA) represent a promising approach to protect agricultural harvest as “natural pesticides”. They are biodegradable and are considered to be less harmful for the ecosystem than chemical pesticides. In their study, Tomita, Kuroda and Narihiro discovered the biological control activity of three known strains (available in the Japan Collection of Microorganisms) by using a culture-independent, metagenomic approach. They analyzed the metagenome of six activated sludge samples and identified three promising BCA candidates by bioinformatic analyses focusing on novel NRPS or NRPS/PKS biosynthetic gene clusters. In vitro cultivation confirmed the antifungal activity against Botrytis cinerea.

This study encompasses a rather low sample size (6 activated sludge samples), and these samples were only taken in Japan. However, even this low sample size led to three positive hits: three strains showed antifungal activity against one phytopathogen. The authors provide a solid proof-of-principle for the identification of novel BCAs, but this proof-of-concept needs to be validated by more examples. I believe that the quality of the manuscript can be further improved by addressing the following points:

Only one phytopathogen was analyzed in this study. Testing the antimicrobial activity against further phytopathogens is required to make a more general judgement on the biological control activity of the three bacterial strains.

No positive control was included in the antifungal assay. It is difficult to estimate how potent the BCAs are, since no positive control is shown. Are there any known BCAs that could be used as positive controls? This might also allow to compare the activity of the different BCAs? What happens in a liquid culture of B. cinerea if the bacterial strains are added?

Minor:

Abstract:

The story is hard to follow because the authors assume a lot of prior knowledge by the reader. They should include a brief definition of biological control agents since PLOS ONE has a broad readership who might not be familiar with this term.

Line 19: leave out “having non-ribosomal peptide synthetase (NRPS)”, this does not fit in here. The NRP is responsible for the biological control activity, the NRPS only indirectly by encoding the NRP. This needs to be explained and might be too detailed for the abstract.

Line 21: leave out “with novel NRPS”

Line 27: abbreviation AS for activated sludge has already been introduced in line 25

Line 28: introduce abbreviation PKS (and NRPS if deleted before as suggested above)

Line 28-30: Please correct the sentence. “most of which were assigned to the phyla…” refers to “assembled bins”, but “precited to be novel” refers to NRPS/PKS-related gene sequences”. Thus, the two sentences cannot be connected by “and”.

Line 30: replace “not highly associated with” by “genetically dissimilar to [known NRPS/PKS gene clusters]”

Introduction:

Line 60-62: Move the sentence “A type I NRPS module mainly consists of …” to line 60 (finish explaining type I NRPSs before starting with type II)

Line 82/83: “metagenomic assembled genomes”, the commonly used term is “metagenome assembled genomes (MAGs)”, so this might be introduced here

Line 86: clusters are found in MAGs

Line 89: we propose (instead of we proposed)?

Results and discussion:

Line 225: explain the abbreviation NaPDoS – natural product domain seeker and add a reference for this

Line 230: replace “were not highly associated with” by “were only remotely related to C-domains of the known sequences”

Line 246: please correct the spelling mistake: Ralstonia

Line 262: NAPDOS  NaPDoS, be consistent

Line 265: please correct the spelling mistake: those

Line 274: please correct the spelling mistake: strains

Conclusion:

Line 329: please correct the spelling mistake: resources

Line 342: please correct the spelling mistake: cultured

Figure 2: top branch, please correct the spelling mistake: cyclization, third branch: epimerization domain

Figure caption of Fig 2: blast  BLAST and NAPDOS  NaPDoS

Figure 3: A more descriptive caption should be included.

- How much material was applied? Was a liquid culture of the bacterial strains applied (OD600?, volume in µl?)?

- How did you prepare the 5 mm plugs of B. cinerea? Please include this information in the Materials and Methods section.

- The growth of B. cinerea seems to be different on the N. dentirificans plate, please comment on this phenomenon.

- Is there any possibility to quantify the biological control activity?

Figure 4: Locus tag? Instead of Locas tag?

6. PLOS authors have the option to publish the peer review history of their article (what does this mean?). If published, this will include your full peer review and any attached files.

Reviewer #1: No

Reviewer #2: No

---

## [Author Response · Author response to Decision Letter 0]

28 Sep 2023

Re.: PONE-D-23-16870 

A small step to discover candidate biological control agents from pre-existing bioresources by using novel nonribosomal peptide synthetases hidden in the activated sludge metagenomes

Shun Tomita1*, Kyohei Kuroda1, Takashi Narihiro1*

1 Bioproduction Research Institute, National Institute of Advanced Industrial Science and Technology (AIST), Toyohira-ku, Sapporo, Hokkaido, Japan

＊Co-corresponding authors e-mail addresses: tomita.s@aist.go.jp (ST), t.narihiro@aist.go.jp (TN).

Response to the Journal requirements:

Response: 

We have refined the manuscript under the PLOS ONE's style requirements.

Response: 

We have provided the additional information about the permission for sludge sampling as follows (p.6: L109-110): “A total of 6 AS samples were collected from a full-scale conventional activated sludge tanks treating municipal sewage wastewater in Kyushu region, Japan with the permission of the administrator when necessary.”

3. Please remove any funding-related text from the manuscript and let us know how you would like to update your Funding Statement.

Response:

We have removed the funding-related text from the manuscript and described them in Funding Statement without any changes. 

Response:

Our sequencing data is already open in the DDBJ Sequence Read Archive database as described in “Deposition of DNA sequence data” in Method section (p.10: L182-184): “The raw sequence and binned metagenome data were deposited into the DDBJ Sequence Read Archive database (DRA015582).”

Response to the Editor Comments

The manuscript is pertaining to a current topic of interest and the field of BGCs is now growing with respect to bioinformatics and genome mining. It is harder to deal definitely to deal with metagenomes and it is thus a challenge. However, the lack of positive control is weak point in the current draft in addition to the product itself not being characterized. Please address all the reviewers’ comments carefully in order to have the manuscript improved. It is saying a strong message but also the wording needs work as well.

Response

We sincerely appreciate the valuable and positive comments for our manuscript from the Editor and Reviewers. As noted by the Editor and Reviewers, we agree that information on the positive control is needed. Then, we selected Bacillus subtilis NBRC 109107 as a known BCA producing cyclic lipopeptide, which are synthesized by NRPS, and additionally performed the antifungal activity test in vitro. The results clearly showed that the antifungal activities of candidates were comparable to that of Bacillus subtilis NBRC 109107 (Table 2 and Fig 4). In addition, we performed the antifungal activity test against two phytopathogenic fungus Colletotrichum plurivorum MAFF 306007 and Verticillium dahliaea MAFF 236075. The results indicated that one of the BCA candidate P. ginsengisoli JCM 30745 showed relatively weak activity compared with Bacillus subtilis, and remaining two candidates did not show the activity against C. plurivorum and V. dahliaea. We have added descriptions of these results to clarify the discussion, and we have carefully revised the manuscript in accordance with the comment and a point-by-point reply to the comments is given as follows.

Response to the Reviewer #1: 

The authors described a small-scale test case to select NRPS or NRPS/PKS clusters from activated sludge shotgun metagenome data. Next, they looked at culture collections for strains that harbor the selected clusters, with the end goal to use these strains as biological control agents. Next, they investigated the antifungal activity of these strains. Even though the authors identified three strains that harbor a similar NRPS gene between them, the claim that the product of this NRPS is responsible for the antifungal activity is weak, and further experiments are needed to corroborate with this hypothesis. Moreover, some sections must be re-written to be more clear about how and why the analysis was performed.

Response

We sincerely appreciate the valuable comments for our manuscript. We have attempted to clarify the issues you raised and have responded to your comments point by point as follows. 

General points:

Abstract: the abstract should be re-written. E.g. (Pg. 8, Ln. 22-26), it is unclear the connection between the culture collection strains and the shotgun metagenome data; (Pg. 9, Ln. 37-41), the conclusion is too vague. The problem persists through the introduction topics.

Response

Thank you for helpful comments. We have deleted the unclear sentence “from cultured strains available from culture collections” and refined the sentence as follows (p.2: L23-24):

“we attempted to explore novel BCA candidates by employing the shotgun metagenomic information of the activated sludge (AS) microbiome, which is thought to contain unutilized biological resources.”

In addition, we have expressed the conclusion clearly as follows (p.3: L38-39):

“Our findings provide a possible strategy for discovering novel BCAs buried in the environment using genome-resolved metagenomics.”

Pg. 14-15, Ln. 135-152/ Pg.19, Ln. 223-231/ Pg. 22, Ln. 260-263: section on C-domain phylogeny approach should be re-written or removed. No information is obtained from this analysis.

Response

We apologize for the lack of clarity in our explanation. We believe that phylogenetic analysis of the C-domain is essential to search the BCA candidate strains in this paper because C domain-based phylogeny has been used to assess the diversity and richness of NRPS-harboring microorganisms in environments and identify novel functional classes that may be associated with uncharacterized biosynthetic mechanisms. (Roongsawang et al. 2005). Hence, to clarify the objective of this analysis, we have made a clear distinction between the AntiSMASH analysis and phylogenetic analysis of the C-domain by modifying the sentences and changing the order of sentences related to the phylogenetic analysis of the C-domain. (p7: L133-134, p15-17: L267-268, L270-307)

Pg. 17, Ln. 197-198: it would be interesting to discuss the NRPS and NRPS/PKS domain architecture in this section. E.g., do they have a complete set of modules, how many?

Response

Thank you for your helpful comments. If you mean “complete set of modules” is a complete NRPS for producing one NRP, then it is not possible to define what constitutes a complete NRPS since most of the compounds detected in this study are unidentified. In addition, it would be very difficult to extract a complete NRPS set since short read shotgun metagenomic sequencing was used in this analysis. Instead, if you mean "complete set of modules" is a set of A, T, and C domains that construct an NRPS module, the complete modules were found in 22 clusters (22.45% of the total), among which are 14 with one complete set of modules, 7 with two sets, and 1 with three sets. Hence, we have added the number of NRPS and NRPS/PKS hybrid with complete modules in Table 1. In addition, we have included the information for discussing the NRPS and NRPS/PKS clusters observed in this study as follows (p.11: L207-213, Table1):

“A total of 78 NRPS (15.2%) and 19 hybrid NRPS/PKS (3.6%) clusters were observed in 43 bins (Fig 1A). The complete modules containing A, T, and C domains were found in 22 clusters (22.45% of the total) (Table 1), among which are 14 with one complete set of modules, 7 with two sets, and 1 with three sets. A previous study reported that NRPS and PKS in the AS MAGs appeared to be mostly very short despite being uninterrupted by contig breaks, suggesting that PKs and NRPs in AS produce simple compounds compared with the structure of the known multimodular NRPS and PKS (Sanchez-navarro et al., 2022)”.

Pg. 23, Ln. 285-300: even though there might be a correlation between the presence of this NRPS with the antifungal activity, the results presented in this section are not enough to claim it. There are many options to follow: 1. perform reverse genetics in one of the bacterial strains and delete the NRPS, and then repeat the antifungal assay. 2. grow at least one of the strains or perform heterologous expression to isolate the NRPS product, then repeat the antifungal assay it the purified molecule. If none of these are done, this section should be toned down.

Response

Thank you for your helpful comments. We agree that additional experiments #1 and/or #2 as you suggested would be valuable and are clearly needed for the specify the gene(s) for NRPS production. Unfortunately, however, because of the difficulty of genetic modifications for non-model organisms (i.e., the genera Pseudoduganella, Noviherbaspirillum, and Rhodoferax), we are unable to succeed the experiments. Therefore, we have stated that these experiments are necessary to identify the exact products for antifungal activity, and toned down the claim the product of this NRPS is responsible for the antifungal activity (p.20: L356-360):

“In the future, to confirm the exact NRPS products with the antifungal activity, it is clearly necessary to investigate in more detail whether this NRPS gene cluster is involved in NRP biosynthesis by constructing a gene disruption mutant or using a heterologous expression system and by identifying the chemical structure using mass spectrometry.”

Pg. 24, Ln. 305-313: it would be more clear if the annotation of genes belonging to the NRPS gene cluster to be discussed before, in the first section of the results.

Response

Thank you for your helpful comments. We have moved the discussion of the NRPS gene cluster to the end of the first section of the results. (p.17: L 296-307). With this modification, we have replaced Fig 4 with Fig 3.

Specific points:

Title: correct nonribosomal peptide synthase to nonribosomal peptide synthetase.

Response

We apologize for the mistake. We have modified it. (p.1: Title)

Pg. 18, Ln. 203: correct 181 to 180.

Response

We apologize for the mistake. We have modified it. (Throughout the manuscript)

Pg. 18, Ln. 215-217: phylum names should not be italicized. Problem persists throughout the whole manuscript.

Response

We apologize for the mistake. We have modified them. (Throughout the manuscript)

Pg. 18, Ln. 217-218: what does it mean NRPS and NRPS/PKS hybrid gene clusters, with each 31-33% proportion? Sentences must be re-written.

Response

We apologize for lack of explanation of the sentence We have rewritten “31-33% proportion” to “33.3% (NRPS) and 31.3% (NRPS/PKS hybrid) proportion” (p.14: L238)

Pg. 19, Ln. 222: replace unused by underexploited.

Response

As you pointed out, we have replaced “unused” by “underexploited”. (p.14: L242)

Pg. 21, Ln. 237: replace Myxococcota are by Myxococcota is.

Response

As you pointed out, we have replaced “Myxococcota are” by “Myxococcota is”. (p.14: L244)

Pg. 21, Ln. 253: replace including by producing.

Response

As you pointed out, we have replaced “including” by “producing”. (p.15: L260)

Pg. 22, Ln. 269-270: genera names should not be abbreviated here.

Response

As you pointed out, we have modified them. (p.17: L293-294)

Pg. 23, Ln. 297: SI figure says Massilia ginsengisoli. Please check the correct genera name.

We apologize for the mistake. We have modified it. (Throughout the manuscript)

Response to the Reviewer #2: 

This study encompasses a rather low sample size (6 activated sludge samples), and these samples were only taken in Japan. However, even this low sample size led to three positive hits: three strains showed antifungal activity against one phytopathogen. The authors provide a solid proof-of-principle for the identification of novel BCAs, but this proof-of-concept needs to be validated by more examples. I believe that the quality of the manuscript can be further improved by addressing the following points: Only one phytopathogen was analyzed in this study. Testing the antimicrobial activity against further phytopathogens is required to make a more general judgement on the biological control activity of the three bacterial strains. No positive control was included in the antifungal assay. It is difficult to estimate how potent the BCAs are, since no positive control is shown. Are there any known BCAs that could be used as positive controls? This might also allow to compare the activity of the different BCAs? What happens in a liquid culture of B. cinerea if the bacterial strains are added?

Response 

Thank you for your helpful comments. As you suggested, we have added Ba. subtilis, which is known to produce NRP (Habe et al., 2017), to the antimicrobial activity data as a positive control in Fig 4, Table 2 and S3 Fig. We also made modifications in the information related to Ba. subtilis NBRC 109107 in the material and methods (p.9: L161-162), results and discussion (p. 19-20: L330-360), and the reference sections (p. 32: L599-601). In addition, we have added two phytopathogens, Verticillium dahlia MAFF 236075 and Colletotrichum plurivorum MAFF 306007 for the antifungal activity test. Then, we have added a new Figure (S3 Fig) and modified the sentence in the abstract (p.2: L35-36), material and methods (p.8: L154-156) and results and discussion sections (p.19-20: L330-360) related to the antifungal activity test. With this modification, we have replaced S3Fig with S4Fig.

In addition, thank you for your comments related to a liquid culture of Bo. cinerea. As requested, we cultured Bo. cinerea solely on a liquid culture PDB medium, but the mycelium of Bo. cinerea aggregated. Hence, we think it might be difficult to evaluate the antifungal activity of the BCA strains against Bo. cinerea by using a liquid culture.

Abstract:

The story is hard to follow because the authors assume a lot of prior knowledge by the reader. They should include a brief definition of biological control agents since PLOS ONE has a broad readership who might not be familiar with this term.

Response

Thank you for your helpful comments. We have added the sentences about the definition of biological control agents. (p.2: L18-20)

“Biological control agents (BCAs), beneficial organisms that reduce the incidence or severity of plant disease, have been expected to be alternatives to replace chemical pesticides worldwide.”

Line 19: leave out “having non-ribosomal peptide synthetase (NRPS)”, this does not fit in here. The NRP is responsible for the biological control activity, the NRPS only indirectly by encoding the NRP. This needs to be explained and might be too detailed for the abstract.

Response

As you pointed out, we have left out the explanation about the NRPS. (p.2: L20-21)

Line 21: leave out “with novel NRPS”

Response

As you pointed out, we have left out the explanation about the NRPS. (p.2: L21)

Line 27: abbreviation AS for activated sludge has already been introduced in line 25

Response

We apologize for the mistake. We have rewritten “activated sludge” to “AS”. (p.2: L26)

Line 28: introduce abbreviation PKS (and NRPS if deleted before as suggested above)

Response

We apologize for lack of explanation of PKS. We have added the sentence related to the abbreviation of NRPS and PKS. (p.2: L27-28)

Line 28-30: Please correct the sentence. “most of which were assigned to the phyla…” refers to “assembled bins”, but “precited to be novel” refers to NRPS/PKS-related gene sequences”. Thus, the two sentences cannot be connected by “and”.

Response

As you pointed out, we have rewritten the sentence. (p.2: L28-31)

Line 30: replace “not highly associated with” by “genetically dissimilar to [known NRPS/PKS gene clusters]”

Response

As you pointed out, we have replaced the sentence. (p.2: L30)

Introduction:

Line 60-62: Move the sentence “A type I NRPS module mainly consists of …” to line 60 (finish explaining type I NRPSs before starting with type II)

Response

As you pointed out, we have moved the sentence about type I to above the type II description. (p.4: L58-59)

Line 82/83: “metagenomic assembled genomes”, the commonly used term is “metagenome assembled genomes (MAGs)”, so this might be introduced here

Response

L93-94: As you pointed out, we have rewritten “metagenomic assembled genomes” to “metagenome assembled genomes (MAGs)”. (p.5: L80-81)

Line 86: clusters are found in MAGs

Response

We have reworded the sentence as “MAGs.” (p.5: L84)

Line 89: we propose (instead of we proposed)?

Response

We have reworded the sentence as “we propose~” (p.5: L87)

Results and discussion:

Line 225: explain the abbreviation NaPDoS – natural product domain seeker and add a reference for this

Response

We apologize for the lack of explanation of the abbreviation of NaPDoS. We had already explained the NaPDoS with the reference in the material and method section, so we have added the description of the abbreviation of NaPDoS there. (p.7: L133-134)

Line 230: replace “were not highly associated with” by “were only remotely related to C-domains of the known sequences”

Response

As you pointed out, we have replaced “were not highly associated with” by “were only remotely related to C-domains of the known sequences.” (p.16: L280)

Line 246: please correct the spelling mistake: Ralstonia

Response

We apologize for the misspelling. We have modified it. (p.15: L253)

Line 262: NAPDOS  NaPDoS, be consistent

Response

We apologize for the misspelling. We have modified it. (Throughout the manuscript)

Line 265: please correct the spelling mistake: those

Response

We apologize for the misspelling. We have modified it. (L16: L288)

Line 274: please correct the spelling mistake: strains

Response

L295: We apologize for the misspelling. We have modified it. (Throughout the manuscript)

Conclusion:

Line 329: please correct the spelling mistake: resources

Response

We apologize for the misspelling. We have modified it. (p.21: L374) 

Line 342: please correct the spelling mistake: cultured

Response

We apologize for the misspelling. We have modified it. (p.21: L387)

Figure 2: top branch, please correct the spelling mistake: cyclization, third branch: epimerization domain

Response

We apologize for the misspelling. We have modified it. (Fig 2)

Figure caption of Fig 2: blast  BLAST and NAPDOS  NaPDoS

Response

We apologize for the misspelling. We have modified it. (p.17: L309-311)

Figure 3: A more descriptive caption should be included.

- How much material was applied? Was a liquid culture of the bacterial strains applied (OD600?, volume in µl?)?

Response

Thank you for your helpful comment. The overnight liquid culture was dipped with the inoculating loop and streaked onto the agar plate, so we could not measure the amount of the inoculated liquid culture. Hence, we have added “Overnight culture of each bacterial strain was inoculated in a straight line on R2A agar plate with an inoculating loop” to the caption of Fig 4. (p.21: L370-371: caption of Fig 4)

- How did you prepare the 5 mm plugs of B. cinerea? Please include this information in the Materials and Methods section.

Response

We apologize for lack of the information in materials and methods. We have added the information related to preparation of the pathogens. (p.9: L168-171)

- The growth of B. cinerea seems to be different on the N. dentirificans plate, please comment on this phenomenon.

Response

We agree that B. cinerea seems to differ from other strains on the N. denitrificans plate. This phenomenon was sometimes observed not only on N. denitirificans plate but also on other bacterial strain plates. Unfortunately, we cannot explain this phenomenon, but we have replaced N. denitrificans photo in Fig.4 with a better one. (Fig 4)

- Is there any possibility to quantify the biological control activity?

Response

Thank you for your helpful comments. As requested, we have added a new table 2 (p. 20), which shows the quantification of the antifungal activity against Bo. cinerea. In addition, we have added sentences related to methods of the statistical analyses and quantification of antifungal activity in the material and methods section. (Table2, p.9-10: L172-180)

Figure 4: Locus tag? Instead of Locas tag?

Response

We apologize for the misspelling. We have modified it. (Fig 3)

---

## [Decision Letter · Decision Letter 1]

15 Oct 2023

PONE-D-23-16870R1A small step to discover candidate biological control agents from pre-existing bioresources by using novel nonribosomal peptide synthetases hidden in the activated sludge metagenomesPLOS ONE

Dear Dr. Tomita,

Thank you for submitting your manuscript to PLOS ONE. After careful consideration, we feel that it has merit but does not fully meet PLOS ONE’s publication criteria as it currently stands. Therefore, we invite you to submit a revised version of the manuscript that addresses the points raised during the review process. The minor revisions required by the reviewer is very crucial for the manuscript to be accepted, alongside the English proofreading.

We look forward to receiving your revised manuscript.

Kind regards,

Laila Adel Ziko

Academic Editor

PLOS ONE

Journal Requirements:

Reviewers' comments:

Reviewer's Responses to Questions

**Comments to the Author**

1. If the authors have adequately addressed your comments raised in a previous round of review and you feel that this manuscript is now acceptable for publication, you may indicate that here to bypass the “Comments to the Author” section, enter your conflict of interest statement in the “Confidential to Editor” section, and submit your "Accept" recommendation.

Reviewer #1: All comments have been addressed

Reviewer #2: (No Response)

2. Is the manuscript technically sound, and do the data support the conclusions?

Reviewer #1: Yes

Reviewer #2: Yes

3. Has the statistical analysis been performed appropriately and rigorously? 

Reviewer #1: N/A

Reviewer #2: I Don't Know

4. Have the authors made all data underlying the findings in their manuscript fully available?

Reviewer #1: Yes

Reviewer #2: Yes

5. Is the manuscript presented in an intelligible fashion and written in standard English?

Reviewer #1: Yes

Reviewer #2: No

6. Review Comments to the Author

Reviewer #1: Authours addressed all previous comments. No other edits are needeed at this point.

Reviewer #2: My previous points were addressed, but unfortunately the manuscript was not written with due care. Strain names were repeatedly misspelled and abbreviated incorrectly (Noviherbaspirillum denitrificans is correct, not dentrificans or denitirificans, B. subtilis not Ba. subtilis, etc.), even in the revised version. I do not consider it my responsibility as a reviewer to correct all these errors, but I strongly recommend that they be corrected before the manuscript is published. The authors should consider having a native speaker proofread the manuscript (in the best case, this should happen before the first submission).

Figure 4:

The authors should mention that R. sediminis JCM 32677 was used as a negative control in the figure caption.

7. PLOS authors have the option to publish the peer review history of their article (what does this mean?). If published, this will include your full peer review and any attached files.

Reviewer #1: No

Reviewer #2: No

---

## [Author Response · Author response to Decision Letter 1]

25 Oct 2023

Response to the Journal requirements:

Response: 

We have reviewed and modified the reference list under the PLOS ONE's style requirements (p23:L408, 413, p24:L438, p25:L443, p26:L464, 467, p29:L529, p30:L543, 547, 551, p31:L559, p32:L584).

 

Response to the Editor and Reviewers

We sincerely appreciate the valuable and positive comments on our revised manuscript from the Editor and Reviewers. A point-by-point response to the comments is given below. And we have added the Figure 4 caption, which did not explain negative control. In addition, as you can see in the attached certificate, we have Elsevier Language Editing Service proofread the manuscript. Please note that the Line numbers in the responses correspond to the revised manuscript with changes marked.

Reviewer comments:  

1.My previous points were addressed, but unfortunately the manuscript was not written with due care. Strain names were repeatedly misspelled and abbreviated incorrectly (Noviherbaspirillum denitrificans is correct, not dentrificans or denitirificans, B. subtilis not Ba. subtilis, etc.), even in the revised version. I do not consider it my responsibility as a reviewer to correct all these errors, but I strongly recommend that they be corrected before the manuscript is published. The authors should consider having a native speaker proofread the manuscript (in the best case, this should happen before the first submission). 

Response

We apologize for the mistakes. We have modified the strain names (Fig 3,4, S3 Fig, and Throughout the manuscript). In addition, the manuscript has been edited by an Elsevier Language Editing Service, as you can see in the attached certificate.

2.Figure 4: The authors should mention that R. sediminis JCM 32677 was used as a negative control in the figure caption.

Response

We apologize for lack of the explanation of negative control. We have added the Figure 4 caption (p21: L370-371).

---

## [Decision Letter · Decision Letter 2]

10 Nov 2023

A small step to discover candidate biological control agents from preexisting bioresources by using novel nonribosomal peptide synthetases hidden in the activated sludge metagenomes

PONE-D-23-16870R2

Dear Dr. Tomita,

We’re pleased to inform you that your manuscript has been judged scientifically suitable for publication and will be formally accepted for publication once it meets all outstanding technical requirements.

Kind regards,

Laila Adel Ziko

Academic Editor

PLOS ONE

Additional Editor Comments (optional):

Reviewers' comments:

Reviewer's Responses to Questions

**Comments to the Author**

1. If the authors have adequately addressed your comments raised in a previous round of review and you feel that this manuscript is now acceptable for publication, you may indicate that here to bypass the “Comments to the Author” section, enter your conflict of interest statement in the “Confidential to Editor” section, and submit your "Accept" recommendation.

Reviewer #2: All comments have been addressed

2. Is the manuscript technically sound, and do the data support the conclusions?

Reviewer #2: (No Response)

3. Has the statistical analysis been performed appropriately and rigorously? 

Reviewer #2: (No Response)

4. Have the authors made all data underlying the findings in their manuscript fully available?

Reviewer #2: (No Response)

5. Is the manuscript presented in an intelligible fashion and written in standard English?

Reviewer #2: (No Response)

6. Review Comments to the Author

Reviewer #2: (No Response)

7. PLOS authors have the option to publish the peer review history of their article (what does this mean?). If published, this will include your full peer review and any attached files.

Reviewer #2: No

---

## [Editor Report · Acceptance letter]

15 Nov 2023

PONE-D-23-16870R2 

A small step to discover candidate biological control agents from preexisting bioresources by using novel nonribosomal peptide synthetases hidden in activated sludge metagenomes 

Dear Dr. Tomita:

I'm pleased to inform you that your manuscript has been deemed suitable for publication in PLOS ONE. Congratulations! Your manuscript is now with our production department. 

Kind regards, 

on behalf of

Dr. Laila Adel Ziko 

Academic Editor

PLOS ONE